# Continuing Medical Education Improves Physician Communication Skills and Increases Likelihood of Pediatric Vaccination: Findings from the Pediatric Influenza Vaccination Optimization Trial (PIVOT)—II

**DOI:** 10.3390/vaccines11010017

**Published:** 2022-12-21

**Authors:** William A. Fisher, Vladimir Gilca, Michelle Murti, Alison Orth, Hartley Garfield, Paul Roumeliotis, Emmanouil Rampakakis, Vivien Brown, John Yaremko, Paul Van Buynder, Constantina Boikos, James A. Mansi

**Affiliations:** 1Department of Psychology, Western University, London, ON N6A 3K7, Canada; 2Department of Social and Preventive Medicine, Québec Public Health Institute, Laval University, Québec City, QC G1V 5B3, Canada; 3Dalla Lana School of Public Health, University of Toronto, Toronto, ON M5T 3M7, Canada; 4Fraser Health Authority, Vancouver, BC V3T 0H1, Canada; 5The Hospital for Sick Children, University of Toronto, Toronto, ON M5G 1X8, Canada; 6Eastern Ontario Health Unit, Cornwall, ON K6J 5T1, Canada; 7JSS Medical Research, Montreal, QC H4S 1N8, Canada; 8Department of Family and Community Medicine, University of Toronto, Toronto, ON M5G 1V7, Canada; 9The Montreal Children’s Hospital, Montreal, QC H4A 3J1, Canada; 10Department of Pediatrics, McGill University, Montreal, QC H3A 0G4, Canada; 11School of Medicine, Griffith University, University of Western Australia, Perth, WA 6009, Australia; 12Seqirus, Montreal, QC H9H 4M7, Canada

**Keywords:** influenza, childhood vaccination, parental acceptance, vaccine hesitancy, education

## Abstract

This study evaluated the impact of a continuing medical education (CME) program that emphasized actionable information, motivation to act, and skills to strengthen physician recommendations for seasonal influenza vaccination in children 6 through 23 months of age for whom influenza immunization rates are suboptimal. Physicians were randomly assigned to an accredited CME program or to no CME. Participants completed pre- and post-study questionnaires. Influenza immunization rates were compared between groups. A total of 33 physicians in the CME group and 35 in the control group documented 292 and 322 healthy baby visits, respectively. Significantly more parents immunized their children against influenza after interacting with CME-trained physicians than those with no CME training (52.9% vs. 40.7%; *p* = 0.007). The odds ratio for vaccination after visits with CME-trained physicians was 1.52 (95% confidence interval 1.09 to 2.12; *p* = 0.014), which was unaffected by the socioeconomic status of parents. Parents who discussed influenza vaccination with CME-trained physicians were 20% more likely to choose an approved but publicly unfunded adjuvanted pediatric influenza vaccine. The percentages of physicians reporting the highest levels of knowledge, ability, and confidence doubled or tripled after the CME intervention. Significantly more parents immunized very young children after interacting with physicians who had undergone CME training.

## 1. Introduction

Each year, approximately 13% of children younger than 5 years, or roughly 90 million worldwide, contract influenza [1,2]. Influenza-related death rates for children ≤5 years range between 2.1 and 23.8 per 100,000 population across the globe, and influenza accounts for 10% of pediatric hospitalizations for respiratory illnesses [3,4]. The highest percentages of hospitalizations and death are among children younger than 2 years [5].

Roughly 10% to 20% of Canadians are infected with influenza during influenza season [6], and large numbers of hospitalizations and deaths are attributable to influenza annually [1]. Although infection rates are highest in children 5–9 years of age, children ≤2 years have the highest rates of serious illness and mortality, along with older adults (≥65 years) and individuals with underlying medical conditions [7]. This pattern of age-related vulnerability to influenza, and related morbidity and mortality, is commonly observed worldwide [3].

Vaccination against influenza remains the most important public health strategy to prevent influenza-associated morbidity and mortality. Infants and children 6 through 23 months of age have a high risk of complications and are considered one of the priority groups for influenza immunization by global national immunization technical advisory groups [7,8,9,10,11]. For these youngest eligible influenza vaccine recipients, physicians may communicate the risk of influenza-related illness and influenza vaccine options to parents as part of routine, healthy baby visits. As such, pediatricians and family physicians can play an important role in seasonal vaccine delivery and are expected to play a major role in seasonal influenza vaccination campaigns. Despite recommendations for high influenza immunization coverage in this age group, seasonal influenza vaccine coverage remains suboptimal in many settings worldwide [3,5,12]. In Canada, for example, only 26.5% of children aged 6 months through 4 years were vaccinated against influenza during the 2016–2017 season [6].

Research has shown that the major drivers of parental acceptance of seasonal influenza vaccination are a doctor’s recommendation, prevention of influenza, and a desire to reduce influenza symptoms [13]. Clinicians may be unaware of the potential severity and complications of influenza infection in children younger than 2 years of age; they may lack skills and confidence in discussing influenza vaccination with parents of infants; and they may consequently neglect to offer vaccine options and to recommend vaccination, all of which may contribute to suboptimal vaccination rates [14]. Commonly identified barriers to parental acceptance of vaccination include a low perceived risk of influenza, the perception that the vaccine causes influenza, perceptions of vaccine inefficacy, and potential side effects of vaccination [13,15,16], all of which may be addressed by healthcare provider education.

Continuing medical education (CME) for physicians may foster awareness of best practices in patient care [17,18,19]. Conventional CME platforms, however, have not been shown to be uniformly successful at changing physician behavior in the practice setting [20]. This study evaluated the impact of an Information–Motivation–Behavioral Skills (IMB) model-based approach to CME that emphasizes the provision of actionable information, motivation to act on this information, and skills development for acting effectively in offering and recommending seasonal influenza vaccination to parents of children aged 6 through 23 months. The CME intervention was created on the basis of the IMB model, a well-validated framework for health behavior change, and targeted previously identified clinician and parent barriers to seasonal influenza vaccination uptake [21]. Our hypothesis was that an IMB model-based CME, saturated with actionable counseling information, motivation to apply it in discussion with parents, and skills for educating parents effectively, would alter physician confidence, increase their likelihood of offering and recommending seasonal influenza vaccination, and increase parental acceptance and influenza vaccine choice. We further hypothesized that a tailored CME that featured information, motivation, and skills development concerning the offering and recommendation of an adjuvanted seasonal influenza vaccine of special relevance for infants would strengthen parental preference for this vaccine despite its approved-yet-unfunded status. The adjuvanted vaccine is a trivalent, seasonal influenza vaccine (Fluad™, Seqirus UK Limited, Maidenhead, UK) that has demonstrated significantly greater immunogenicity and efficacy compared to standard, nonadjuvanted vaccines, in infants 6 through 23 months of age.

## 2. Materials and Methods

### 2.1. Study Design and Population

In this multicenter, randomized trial, physicians were recruited from family and general medicine practices in Ontario and randomly assigned in a 1:1 fashion to one of two study arms. One study group received an accredited, IMB model-based, targeted pediatric influenza CME program, and the other study group did not receive the CME and provided routine infant care (Figure 1).

Physician randomization was stratified by regional socioeconomic status to ensure a comparable distribution in the two study groups. The socioeconomic status (high, medium, low) of each practice location was ascertained using income distribution (average household income), neighborhood environment (average market value of dwelling), and educational attainment (percentage of people with a university/graduate degree). Data were obtained from the Statistics Canada 2010 Census. Socioeconomic markers were summed to create a socioeconomic status index, and the resulting distribution was divided into three equal percentiles corresponding to categories of low, medium, and high, which were linked to the first three numbers of the practice location postal code.

Physicians were eligible to participate in the study if they saw ≥10 infants aged 6 through 23 months per month for healthy baby visits.

### 2.2. Study Intervention: IMB Model-Based CME

Physicians randomly assigned to the CME group participated in an online self-learning program, which was certified by the College of Family Physicians of Canada for up to 1 Mainpro+ Credit. The targeted CME intervention emphasized immunization information that was easy to translate into action and was to be shared with parents, motivational content to incline the physician to communicate this information, and behavioral skills coaching illustrating how to offer and recommend influenza vaccination in brief and effective discussion with parents. The online CME took approximately 30–40 min to complete and, though differing substantially in content focus, was no more demanding than more standard CME approaches. A key feature of the tailored CME was the inclusion of video case-based vignettes illustrating a brief, effective, and presumptive offering of pediatric seasonal influenza vaccine and brief, clear, and respectful responses to common parental questions and reservations.

### 2.3. Study Procedures

Physicians in both study groups were required to interact with parents of ≥10 different patients 6 through 23 months of age who were eligible for seasonal influenza vaccination. Each physician–patient interaction occurred during a scheduled healthy baby visit in which all physicians followed the standardized visit flow as per the Rourke Baby Record Guide II, III, or IV, depending on the infant’s age [22,23].

Participating physicians’ knowledge and practices concerning influenza vaccination were assessed via questionnaires before, during, and after the study. All participating physicians (regardless of study group) were required to complete all questionnaires, which took 2–5 min each and could be completed online or on paper and faxed to the data collection center. To establish a baseline, the pre-study questionnaire asked a series of closed and open-ended questions through which physicians described their current immunization practice, their knowledge of influenza, and their ability and confidence in discussing influenza vaccination with parents. After each of the 10 routine healthy baby visits included in the study, physicians completed a questionnaire that described their interaction with each parent regarding seasonal influenza and the parent’s resulting decision on immunization. Following completion of all 10 heathy baby visits, physicians described their post-study knowledge of influenza and their ability and confidence in discussing influenza vaccination with parents in the post-study questionnaire. After completing 10 patient interactions, physicians in the no-CME, routine care group were offered the CME program.

Available vaccines included all approved nonadjuvanted trivalent inactivated influenza vaccines (TIV) and one adjuvanted trivalent inactivated influenza vaccine (aTIV; Fluad^™^, Seqirus UK Limited). The choice of adjuvanted or nonadjuvanted vaccine was made by the parent after discussion with the physician. Parents who chose aTIV, an approved but not publicly funded product, paid for this vaccine.

### 2.4. Statistical Methods

Data analysis was primarily descriptive with measures of central tendency (mean, median) and dispersion (standard deviation, 95% confidence interval [CI] of the mean, range) used for continuous variables and frequency distributions for categorical variables. Key covariates at baseline were compared between the two groups of physicians (Table 1). Pre- and post-study influenza knowledge, ability, and confidence in discussing influenza with parents; strength of vaccine recommendation; and success in recommending influenza vaccination were also evaluated with Chi-squared tests. In addition, logistic regression was used to compare the proportions of influenza immunizations in the two groups while adjusting for socioeconomic status. In this logit model, the cluster effect, i.e., the physician site, was entered as a random effect.

## 3. Results

### 3.1. Baseline Demographics and Immunization Practices

A total of 68 physicians participated in the study, 33 of whom were randomized to the CME intervention group and 35 to the no-CME, routine care group. The majority were family medicine physicians, and only 1.5% had a pediatric focus (Table 1). Overall, physicians had been practicing within their specialty for a median of 13 years (range 1–47 years). The majority (57.4%) practiced in geographic areas with high socioeconomic status; per protocol, randomization was stratified by socioeconomic status, and there was no difference in socioeconomic distribution between study groups (Table 1). Other baseline demographics were also similar between the study groups (Table 1).

Across both study arms, the vast majority of routine pediatric vaccines of any kind were publicly funded. Participating physicians reported giving the seasonal influenza vaccine to only 26.3% of infants younger than 2 years during the previous influenza season, although 89.7% reported at baseline that they routinely discussed seasonal influenza immunization with parents. There were no significant differences between study groups in these behaviors (Table 1).

### 3.2. Influenza Knowledge and Ability and Confidence in Influenza Discussions

At baseline, the majority of physician study participants believed they were at least somewhat (“slightly”, “moderately”, or “very”) well-informed about influenza epidemiology, outcomes, and vaccines, across study groups (Figure 2a). Furthermore, ≥60% of physicians expressed feeling at least somewhat (“slightly”, “moderately”, or “very”) confident in their ability to discuss influenza vaccines with parents prior to randomization (Figure 2b). More than 50% of physicians across study arms reported that it was at least somewhat (“slightly”, “moderately”, or “very”) easy to discuss influenza with parents of children younger than 2 years. At the same time, however, a non-trivial ~30% of physicians said it was at least somewhat (“slightly”, “moderately”, or “very”) difficult to have such discussions with parents of their pediatric patients. Of note, 33.3% of CME group physicians and 37.1% of control group physicians reported some (“very”, “moderately”, or “slightly”) degree of difficulty in discussing influenza vaccines with parents, and 27.3% and 34.3% of the CME and control groups, respectively, were less than fully confident in their discussions with parents. Baseline differences between study groups were not statistically significant.

At the end of the study, more physicians from both groups reported higher degrees of knowledge, ability, and confidence regarding seasonal influenza and influenza vaccines (Figure 2). Notably, in the CME group, the percentages of physicians reporting the highest levels of knowledge, ability, and confidence doubled or tripled after the CME intervention. Paradoxically, the proportions of physicians who reported being not well-informed or having greater difficulty or less confidence in vaccine discussions with parents also increased in the CME group. Between-group differences in post-study knowledge of influenza vaccines (*p* = 0.030, Figure 2a) and confidence in discussing influenza (*p* = 0.006, Figure 2b) were statistically significant.

### 3.3. Healthy Baby Visit Outcomes

A total of 292 healthy baby visits were documented by CME group physicians and 322 by non-CME group physicians. Overall, 95.4% of visits included in the study were scheduled as routine healthy baby visits according to the Rourke schedule, with no difference between study groups. In addition, 22.9% vs. 13.7% of these healthy baby visits were designated by CME and non-CME physicians, respectively, as being “specifically for an influenza vaccination” (*p* = 0.002). Routine pediatric vaccines (e.g., measles-mumps-rubella, varicella, etc.) were administered during 50.0% of the healthy baby visits that occurred during the study (no difference between study groups).

Across both study groups, the topic of seasonal influenza vaccination was raised more often by the physician (87.9%) than the parent (8.1%), and influenza vaccination was recommended by physicians 96.0% of the time (no significant differences between study groups). Physicians in both study groups “strongly” or “moderately strongly” recommended influenza vaccination to parents at similar rates (91.6% and 90.8% of visits with CME and control physicians, respectively). A very small number of physicians in the CME group (4.7%) also discouraged vaccination more often. Additionally, parents decided to immunize children against influenza at the current visit significantly more often after discussions with the CME group physicians (52.9%) compared to those who discussed vaccination with a non-CME physician (40.7%), representing a roughly 30% increase in the CME over the non-CME group (*p* = 0.007). Correspondingly, fewer parents who discussed influenza immunization with a CME-trained physician decided to consider vaccination at a later visit (17.8%) compared to those who discussed vaccination with a non-CME physician (25.4%). As shown in Table 2, the adjusted odds ratio for influenza immunization after discussion with a CME group physician was 1.52 (95% CI 1.09 to 2.12; *p* = 0.014). Socioeconomic status had no effect on the odds of immunization (Table 2).

Significant differences in the choice of vaccine were observed between the study groups (*p* < 0.001). The majority of the visits conducted by physicians in the CME group resulted in the administration of Fluad (aTIV; 57.5%; *n* = 84), Fluzone (19.9%, *n* = 29), and Flulaval (11.0%; *n* = 16), whereas Fluad (48.1%; *n* = 64), Fluviral (29.3%; *n* = 39), and Fluzone (11.3%; *n* = 15) were mainly administered during visits conducted by physicians in the control group. Parents who chose to have their child receive aTIV cited reasons based on greater efficacy and immunogenicity of this vaccine in very young children.

## 4. Discussion

Children aged 6 through 23 months are particularly vulnerable to seasonal influenza infection and associated sequelae, and vaccination coverage in this age range is suboptimal in many settings worldwide [3,5,12,24]. The current research was influenced by the view that targeted CME, created on the basis of a well-validated model of health behavior change [21], would prepare physicians to effectively communicate about seasonal influenza vaccination for infants and be superior to usual care in relation to completed vaccinations. The data we have collected largely confirm this view. Physicians who received the CME focused on information, motivation, and behavioral skills doubled or tripled their assessments of knowledge, ability, and confidence in the pediatric seasonal influenza vaccine area, and they increased completed vaccinations by a statistically significant 30% (or 50% in the adjusted analysis). Findings that support a 30% to 50% increase in existing suboptimal seasonal influenza vaccination rates in infants would appear to have clinical and public health significance. What is more, CME-trained physicians had vaccine-related discussions with parents that more often resulted in parental choice of an adjuvanted but not yet publicly funded vaccine that has demonstrated superior efficacy in the pediatric population [25,26]. Our findings support the utility of theory-based CME that informs, motivates, and coaches skill development that result in measurable pediatric immunization outcomes.

Paradoxically, some physicians in the CME group also reported feeling less well-informed and less confident concerning vaccine discussions in the post-study survey than in the pre-study survey. It may have been the case that the targeted CME encouraged in-depth discussions with parents that proved more difficult to handle and challenged physicians’ sense of how well-informed and confident they really were. Encountering influenza information in the CME might also have stimulated hesitancy in the very small percentage of CME-trained physicians who discouraged parents from vaccinating their infants. Further quantitative and especially qualitative research is needed to verify this speculation.

A potential study limitation involves the fact that our CME training vaccination gains were observed in comparison to a standard-of-care no-CME condition. However, an independent body of evidence demonstrates that information–motivation–skills-focused intervention content is distinctly linked with health behavior change in multiple health domains [27,28]. Future research can explore the utility of IMB model-based CME approaches in preparing clinicians to introduce, offer, and recommend new vaccine formulations in diverse settings and age groups.

## 5. Conclusions

Completion of an IMB model-based CME strengthened physician knowledge and confidence concerning influenza vaccination discussion with parents of infants. Parents seen by physicians in the CME group were approximately 30% to 50% more likely to accept influenza vaccination. Moreover, they were approximately 20% more likely to choose an approved-yet-unfunded influenza vaccine (aTIV), which has demonstrated increased efficacy and immunogenicity in children. This research provides evidence for including CME as an integral component of public healthcare strategies for increasing influenza vaccine acceptance.

## Figures and Tables

**Figure 1 vaccines-11-00017-f001:**
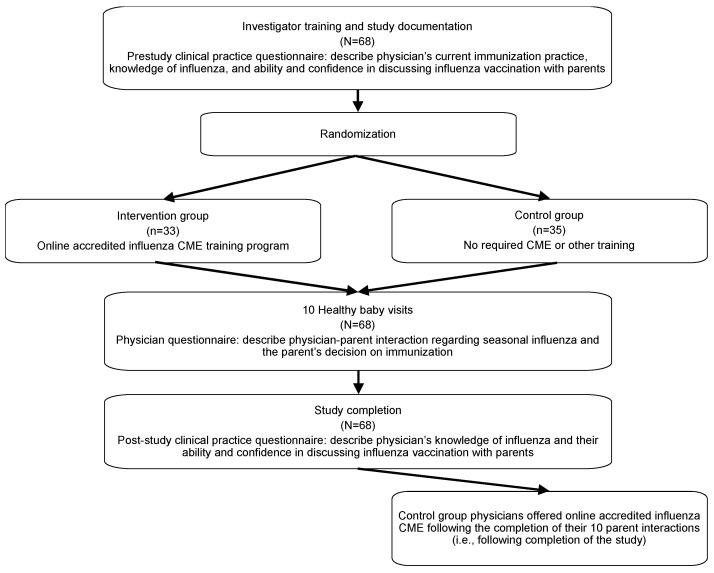
Flow chart of study enrolment and protocol.

**Figure 2 vaccines-11-00017-f002:**
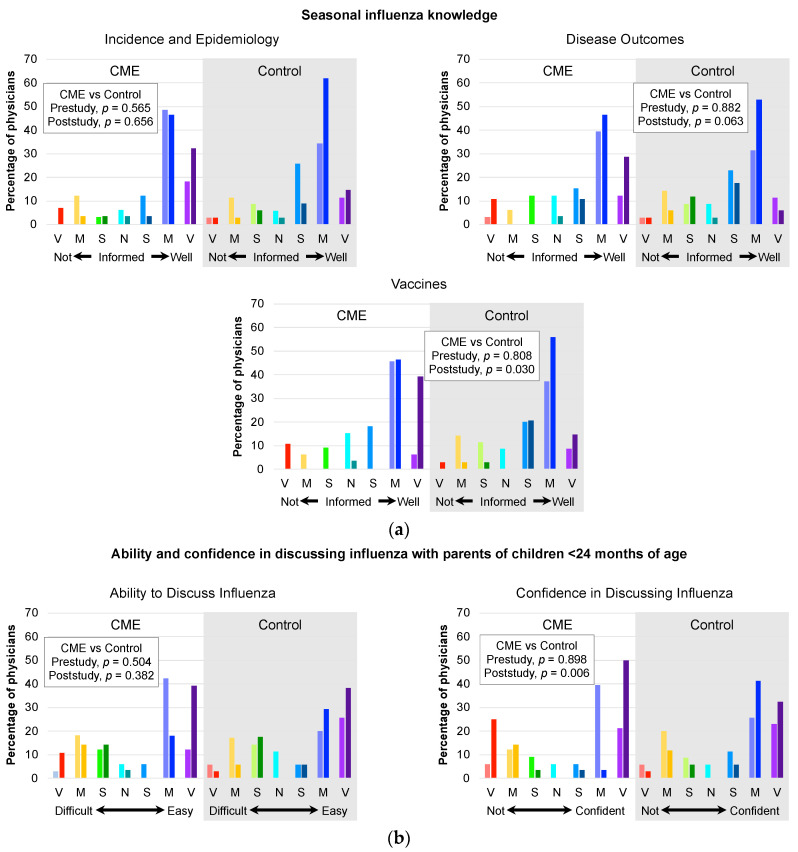
Results of pre- and post-study questionnaire regarding physicians’ knowledge, ability, and confidence in discussing seasonal influenza with parents of children younger than 24 months. (**a**) Seasonal influenza knowledge. Top left question: “How well informed do you feel on seasonal influenza incidence and epidemiology?” Top right question: “How well informed do you feel on the outcomes resulting from pediatric seasonal influenza disease?” Bottom question: “How well informed do you feel on pediatric seasonal influenza vaccines?” (**b**) Left question: “How easy or difficult is it for you to discuss influenza vaccines with parents?” Right question: “What is your level of confidence in discussing influenza vaccines with parents?” For each color pairing, the lighter shade represents the pre-study data and the darker shade represents the post-study data. P values represent comparisons between study groups on all answers to each question and were assessed with Pearson Chi-square. V, very; M, moderately; S, slightly; N, neither.

**Table 1 vaccines-11-00017-t001:** Mixed model-binary logistic regression for the odds of immunization following a physician–parent interaction.

Characteristic	CME Intervention	Control (No CME)	Total	*p* Value ^1^
	(*n* = 33)	(*n* = 35)	(*n* = 68)	
Female sex, *n* (%)	16 (48.5)	21 (60.0)	37 (54.4)	0.465
Therapeutic specialty, *n* (%)				
Family medicine with pediatric focus	0 (0)	1 (2.9)	1 (1.5)	0.222
Family medicine	31 (93.9)	28 (80.0)	59 (86.8)	
General practice	1 (3.0)	5 (14.3)	6 (8.8	
Type of practice, *n* (%)				
Solo	11 (33.3)	14 (40.0)	25 (36.8)	0.621
Group	22 (66.7)	21 (60.0)	43 (63.2)	
Years in practice				
Mean ± SD	18.0 ± 13.2	15.6 ± 10.3	16.8 ± 11.8	0.654
Median (range)	15.0 (1–47)	13.0 (2–40)	13.0 (1–47)	
Days per week in active clinical practice				
Mean ± SD	5.0 ± 0.8	4.9 ± 1.1	4.9 ± 1.0	0.815
Median (range)	5.0 (3-6)	5.0 (2–6)	5.0 (2–6)	
Number of patients seen in practice				
Mean ± SD	1769 ± 850	1856 ± 824	1814 ± 831	0.499
Median (range)	1500 (200–4000)	1700 (200–4009)	1600 (200–4009)	
Percentage of infants 6 through 23 months seen for healthy baby visits per month				
Mean ± SD	14.2 ± 20.6	17.2 ± 21.3	15.8 ± 20.8	0.714
Median (range)	10.0 (1–90)	10.0 (1–80)	10.0 (1–90)	
Approximate percentage of patients 6 through 11 months of age				
Mean ± SD	6.4 ± 6.5	5.7 ± 5.9	6.0 ± 6.2	0.707
Median (range)	5.0 (1–20)	5.0 (1–20)	5.0 (1–20)	
Approximate percentage of patients 12 through 23 months of age				
Mean ± SD	6.8 ± 5.9	6.3 ± 6.0	6.5 ± 5.9	0.506
Median (range)	5.0 (1–20)	5.0 (1–20)	5.0 (1–20)	
Socioeconomic status of physician practice region				
Low, *n* (%)	2 (6.1)	2 (5.7)	4 (5.9)	0.997
Moderate, *n* (%)	12 (36.4)	13 (37.1)	25 (36.8)	
High, *n* (%)	19 (57.6)	20 (57.1)	39 (57.4)	
Percentage of children aged 6 through 23 months vaccinated in practice with routine pediatric vaccines				
Publicly funded vaccines per month, mean ± SD	67.9 ± 39.6	67.1 ± 40.7	67.5 ± 39.8	0.475
Privately paid vaccines per month, mean ± SD	3.4 ± 6.3	4.7 ± 6.5	4.0 ± 6.4	0.807
Percentage of children aged 6 through 23 months vaccinated in practice with the seasonal influenza vaccine during last influenza season, mean ± SD	26.2 ± 20.6	26.4 ± 24.3	26.3 ± 22.4	0.711
Immunization management practices				
Stock and administer vaccines in the clinic, *n* (%)	31 (93.9)	34 (97.1)	65 (95.6)	0.583
Typically initiate discussions with parents on routine pediatric vaccinations, *n* (%)	33 (100)	38 (100)	68 (100)	NA
Typically initiate discussions with parents on seasonal influenza vaccinations, *n* (%)	28 (84.8)	33 (94.3)	61 (89.7)	0.252

^1^ CME vs control, assessed with non-parametric Mann–Whitney test for continuous variables and the Chi-square test for categorical variables. Abbreviations: CME, continuing medical education; NA, not applicable; SD, standard deviation.

**Table 2 vaccines-11-00017-t002:** Mixed model–binary logistic regression for the odds of immunization following a physician parent interaction ^1^.

Covariate	Beta ± SE	OR (95% CI)	*p* Value
Intercept	−0.103 ± 0.207	0.90 (0.60 to 1.36)	0.620
Exposure			
Control group	Ref.	—	—
CME group	0.418 ± 0.170	1.52 (1.09 to 2.12)	0.014
Socioeconomic status			
Low	0.467 ± 0.416	1.60 (0.71 to 3.61)	0.261
Medium	0.136 ± 0.177	1.15 (0.81 to 1.62)	0.440
High	Ref.	—	—

^1^ Physician site was included in the model as a random effect. Abbreviations: CI, confidence interval; OR, odds ratio; SE, standard error.

## Data Availability

Data available upon request from the authors.

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
