# Peer review of "Continuing Medical Education Improves Physician Communication Skills and Increases Likelihood of Pediatric Vaccination: Findings from the Pediatric Influenza Vaccination Optimization Trial (PIVOT)—II"

_vaccines, 2022, doi:10.3390/vaccines11010017_

Round 1

Reviewer 1 Report

The topic is very interesting and the design is ok. However, there are also some parts which need to be improved:

1. it should be a literature review to evaluate the literature about this issue and the results of them.

2. there is an error of the Table Number 1 which can not be found in the manuscript, and there are two Table 2.

3. It will be better for this manuscript if the authors can provide some policy implication and suggestions for medical management department of the government or those related doctors.

Author Response

REVIEWER #1 COMMENTS

AUTHOR RESPONSE

1.     The topic is very interesting and the design is ok.

Thank you for your assessment.

2.     it should be a literature review to evaluate the literature about this issue and the results of them.

A referenced review (16 references) is provided in the Introduction and full references are provided in the bibliography – these were utilized to support the study design stragegy. This review describes the incidence of influenza and pediatric complications for respiratory incidence, (globally and regionally in Canada) the role of vaccines as a public health strategy, the major drivers of parental acceptance of seasonal influenza vaccination, and how continuing medical education may foster best practices in patient care:

3.         Iuliano, A.D.; Roguski, K.M.; Chang, H.H.; Muscatello, D.J.; Palekar, R.; Tempia, S.; Cohen, C.; Gran, J.M.; Schanzer, D.; Cowling, B.J.; et al. Estimates of global seasonal influenza-associated respiratory mortality: a modelling study. Lancet 2018, 391, 1285-1300, doi:10.1016/s0140-6736(17)33293-2.

4.         Lafond, K.E.; Nair, H.; Rasooly, M.H.; Valente, F.; Booy, R.; Rahman, M.; Kitsutani, P.; Yu, H.; Guzman, G.; Coulibaly, D.; et al. Global role and burden of influenza in pediatric respiratory hospitalizations, 1982-2012: a systematic analysis. PLoS medicine 2016, 13, e1001977, doi:10.1371/journal.pmed.1001977.

5.         Poehling, K.A.; Edwards, K.M.; Griffin, M.R.; Szilagyi, P.G.; Staat, M.A.; Iwane, M.K.; Snively, B.M.; Suerken, C.K.; Hall, C.B.; Weinberg, G.A.; et al. The burden of influenza in young children, 2004-2009. Pediatrics 2013, 131, 207-216, doi:10.1542/peds.2012-1255.

6.         Public Health Agency of Canada. 2016/17 Seasonal influenza vaccine coverage in Canada. 2018.

7.         Public Health Agency of Canada. Canadian Immunization Guide Chapter on Influenza and Statement on Seasonal Influenza Vaccine for 2017–2018. Available online: https://www.canada.ca/en/public-health/services/publications/healthy-living/canadian-immunization-guide-statement-seasonal-influenza-vaccine-2017-2018.html. (accessed on

8.         World Health Organization. Influenza (seasonal). Available online: https://www.who.int/en/news-room/fact-sheets/detail/influenza-(seasonal) (accessed on 19 April).

9.         Australian Government Department of Health. 2019 Influenza vaccines. Available online: https://beta.health.gov.au/news-and-events/media-releases/2019-influenza-vaccines (accessed on 19 April).

10.       Grohskopf, L.A.; Sokolow, L.Z.; Broder, K.R.; Walter, E.B.; Fry, A.M.; Jernigan, D.B. Prevention and Control of Seasonal Influenza with Vaccines: Recommendations of the Advisory Committee on Immunization Practices-United States, 2018-19 Influenza Season. MMWR Recomm Rep 2018, 67, 1-20, doi:10.15585/mmwr.rr6703a1.

11.       Joint Committe on Vaccination and Immunisation. Advice on influenza vaccines for 2019/20 Available online: https://www.gov.uk/government/groups/joint-committee-on-vaccination-and-immunisation#influenza-vaccines-jcvi-advice (accessed on 19 April).

12.       Campitelli, M.A.; Inoue, M.; Calzavara, A.J.; Kwong, J.C.; Guttmann, A. Low rates of influenza immunization in young children under Ontario's universal influenza immunization program. Pediatrics 2012, 129, e1421-1430, doi:10.1542/peds.2011-2441.

13.       Flood, E.M.; Rousculp, M.D.; Ryan, K.J.; Beusterien, K.M.; Divino, V.M.; Toback, S.L.; Sasane, M.; Block, S.L.; Hall, M.C.; Mahadevia, P.J. Parents' decision-making regarding vaccinating their children against influenza: A web-based survey. Clin Ther 2010, 32, 1448-1467, doi:10.1016/j.clinthera.2010.06.020.

14.       Dominguez, S.R.; Daum, R.S. Physician knowledge and perspectives regarding influenza and influenza vaccination. Human vaccines 2005, 1, 74-79.

15.       Smith, L.E.; Webster, R.K.; Weinman, J.; Amlot, R.; Yiend, J.; Rubin, G.J. Psychological factors associated with uptake of the childhood influenza vaccine and perception of post-vaccination side-effects: A cross-sectional survey in England. Vaccine 2017, 35, 1936-1945, doi:10.1016/j.vaccine.2017.02.031.

16.       Larson, H.J.; Jarrett, C.; Eckersberger, E.; Smith, D.M.; Paterson, P. Understanding vaccine hesitancy around vaccines and vaccination from a global perspective: a systematic review of published literature, 2007-2012. Vaccine 2014, 32, 2150-2159, doi:10.1016/j.vaccine.2014.01.081.

17.       Bloom, B.S. Effects of continuing medical education on improving physician clinical care and patient health: a review of systematic reviews. International journal of technology assessment in health care 2005, 21, 380-385.

18.       Davis, D.; Evans, M.; Jadad, A.; Perrier, L.; Rath, D.; Ryan, D.; Sibbald, G.; Straus, S.; Rappolt, S.; Wowk, M.; et al. The case for knowledge translation: shortening the journey from evidence to effect. Bmj 2003, 327, 33-35, doi:10.1136/bmj.327.7405.33.

19.       Grimshaw, J.M.; Eccles, M.P.; Lavis, J.N.; Hill, S.J.; Squires, J.E. Knowledge translation of research findings. Implementation science : IS 2012, 7, 50, doi:10.1186/1748-5908-7-50.

3.     there is an error of the Table Number 1 which can not be found in the manuscript, and there are two Table 2.

Thank you for calling our attention to the table numbering. This has been corrected and labels revised accordingly…

4.     It will be better for this manuscript if the authors can provide some policy implication and suggestions for medical management department of the government or those related doctors.

Thank you for this excellent suggestion. The authors have included a public health care statement in the Conclusion section of the manuscript, “This research provides evidence for including CME as an integral component of public health care strategies for increasing influenza vaccine acceptance.”

Author Response

REVIEWER #2 COMMENTS

AUTHOR RESPONSE

The manuscript details an experiment on introducing a new type of continuing medical education (CME) based on an Information-Motivation-Behavioral (1MB) model. The subjects of the study are pediatricians working with infants under two years old. The goal of the study is to evaluate the effect of the IMB-CME on the pediatricians' interaction with parents regarding the necessity of influenza vaccination for infants, and the effects of the interaction on the vaccination rates. The problem of chronic influenza under vaccination stays relevant in many nations across the globe. If the significant positive effect of the IMB-CME approach can be established in the current (and similar) studies and followed by its implementation into the CME routine, it will greatly benefit mankind. The authors chose a solid methodological approach and were able to carry it out. The reviewer was pleasantly surprised with the well-designed experiment with a control group.

Thank you for your supportive assessment.

1.     In the introduction the authors state that they want to test the hypothesis that IMB-CME motivates doctors for action and discussion about the vaccination, etc. In other words, IMB-CME increases physicians' confidence, knowledge, and ability to discuss the topic. The same can be found in the discussion and conclusions sections. However, the statistical tests done in the paper do not test this hypothesis. The Chi-squared test tests for a difference between distribution of two populations, not that one distribution is "bigger" than the other (unlike, for example, one-sided variation of Mann-Whitney test for one distribution being stochasticaly greater than the other). The linear regression analysis shows that IMB-CME increases odds of vaccination, not an increase of confidence or knowledge in physicians. The authors should either rephrase their hypothesis or do a different test in section 3.2. If the authors want to rephrase their hypothesis, the reviewer recommends focusing on results in section 3.3 about the increase in vaccination rates. If the authors are willing to do the test appropriate for their hypothesis, the reviewer suggests ordinal regression or Cochran-Armitage Test.

A)   The authors state the Data analysis was primarily descriptive with measures of central tendency (mean, median) and dispersion (standard deviation, 95% confidence interval [CI] of the mean, range) used for continuous variables and frequency distributions for categorical variables. Key covariates at baseline were compared between the two groups of physicians (Table 1).

B)    Further, logistic regression was used to compare the proportions of influenza immunizations in the two groups while adjusting for socioeconomic status. In this logit model, the cluster effect, i.e., the physician site, was entered as a random effect.

The results are presented in accordinance with the methodology employed.

C)    line 47: could not find any mentioning of influenza in reference 2

Refers to the under 5 mortality rate found in the metadata globally.

D)   line 52: reference 6 is a report on vaccination and reasoning for or against vaccination. No mentioning of infection rates. However, the manuscript states "10% to 20% of Canadians are infected with influenza during influenza season" and cites the report.

E)    lines 174 - 175: table number and caption are for Table 2. Should be for Table 1

Thank you this has been corrected.

F)    Table 1 would benefit from a little formatting. E.g., adding horizontal lines or additional spacing between the groups

Thank you. Table characteristic headers have been reformatted bold to assist the reader. The table format is provided by the journal.

G)   Table 1: first row, column "Control (no CME)" n should be equal to 35, not 38.

This has been corrected.

H)   Figure 3 is missing

This figure was eliminated from the manuscript, and text has been corrected to remove the reference.

I)      line 351: reference 7 text is incomplete after the link

This has been corrected.

J)      There are newer reports for references 2, 6, 7, 9, and 11 on the corresponding websites

These are referenced in accordance with the timeframe the research was conducted.

K)   The study states how well-supported the IMB approach is, however all references (21, 27, and 28) focus on cases with HIV patients. It might be a good idea to add a few more references that describe other cases besides HIV to illustrate generalizability of the methodology.

Round 2

Reviewer 1 Report

This version looks much better.

However, the regression model still needs improved, since the Mixed model–binary logistic regression only controlled one variable-SES. Actrually, the dependent variable is also affacted by other variables, such as doctors' age, medical experiences (years), their fields of study(spcialization), which also should be put into the model as controlling variables.

Author Response

REVIEWER COMMENTS

AUTHOR RESPONSE

1.      "This version looks much better.:

However, the regression model still needs improved, since the Mixed
model–binary logistic regression only controlled one variable-SES.
Actrually, the dependent variable is also affacted by other variables,
such as doctors' age, medical experiences (years), their fields of
study(spcialization), which also should be put into the model as
controlling variables."

a)      Because this was a randomized trial, stratified by socioeconomic status of clinical setting (physician) and controlling for cluster effects, we expected—and achieved--equivalent distributions of all relevant variables in the two groups of IMB CME trained and comparator physicians. As can be seen in Table 1, this expectation was completely empirically confirmed: the two groups of physicians are equivalent on all dimensions thus reassuring us that inequivant variables did not affect outcomes and that the sole salient difference between the two conditions was the IMB CME intervention. Accordingly, the regression analysis appropriately controlled for SES of parents, a variable that could have influenced one of the outcome measures—choosing and paying for aTIV vaccine.

b)     Regarding the study hypothesis, the Authors would like to note that we have tested this specific hypothesis in two separate analyses. First, we report that our analysis shows that “The percentages of physicians reporting the highest levels of knowledge, ability, and confidence doubled or tripled after the CME intervention.” Thus, we confirm empirically our hypothesis that the CME intervention “increases physicians' confidence, knowledge, and ability to discuss the topic.”  Second, we report that “The odds ratio for vaccination after visits with CME-trained physicians was 1.52 (95% confidence interval 1.09 to 2.12; P=0.014), which was unaffected by socioeconomic status of parents” and that “Parents who discussed influenza vaccination with CME-trained physicians were 20% more likely to choose an approved but publicly unfunded adjuvanted pediatric influenza vaccine,” thus empirically confirming that the CME intervention would “increase parental acceptance and influenza vaccine choice.” We note that our discussion of CME intervention impact explicitly cites these statistical outcomes.

c)      [This is our hypothesis text for reference]. Our hypothesis was that an IMB model-based CME, saturated with actionable counselling information, motivation to apply it in discussion with parents, and skills for educating parents effectively, would alter physician confidence, increase their likelihood of offering and recommending seasonal influenza vaccination, and increase parental acceptance and influenza vaccine choice. We further hypothesized that a tailored CME that featured information, motivation, and skills development concerning the offering and recommendation of an adjuvanted seasonal influenza vaccine of special relevance for infants would strengthen parental preference for this vaccine despite its ap-proved-yet-unfunded status. The adjuvanted vaccine is a trivalent, seasonal influenza vaccine (Fluad™, Seqirus UK Limited) that has demonstrated significantly greater immunogenicity and efficacy compared to standard, nonadjuvanted vaccines, in infants 6 through 23 months of age.